# Neurons in the *Nucleus papilio* contribute to the control of eye movements during REM sleep

C. Gutierrez Herrera [1,2,4], F. Girard[3,4]*, A. Bilella [3], T.C. Gent[1], D.M. Roccaro-Waldmeyer [3], A. Adamantidis[1,2] & M.R. Celio[3]*

Rapid eye movements (REM) are characteristic of the eponymous phase of sleep, yet the underlying motor commands remain an enigma. Here, we identified a cluster of Calbindin-D28K-expressing neurons in the *Nucleus papilio* (NP$^{Calb}$), located in the dorsal para-gigantocellular nucleus, which are active during REM sleep and project to the three contralateral eye-muscle nuclei. The firing of opto-tagged NP$^{Calb}$ neurons is augmented prior to the onset of eye movements during REM sleep. Optogenetic activation of NP$^{Calb}$ neurons triggers eye movements selectively during REM sleep, while their genetic ablation or optogenetic silencing suppresses them. None of these perturbations led to a change in the duration of REM sleep episodes. Our study provides the first evidence for a brainstem premotor command contributing to the control of eye movements selectively during REM sleep in the mammalian brain.

---

[1] Center for Experimental Neurology, Department of Neurology, Inselspital, University of Bern, Freiburgstrasse 3, CH-3010 Bern, Switzerland. [2] Department of Biomedical Research (DBMR), University of Bern, Murtenstrasse 50, CH-3008 Bern, Switzerland. [3] Anatomy and Program in Neuroscience, Faculty of Science and Medicine, University of Fribourg, Rte. Albert Gockel 1, CH-1700 Fribourg, Switzerland. [4] These authors contributed equally: C. Gutierrez Herrera, F. Girard. *email: franck.girard@unifr.ch; marco.celio@unifr.ch

Sleep is characterised by rapid transitions between REM (rapid eye movement) and non-REM states, the former being hallmarked by rapid eye movements (EMs), atony of the postural muscles, desynchronized electro-encephalographic (EEG) activity and vivid dreaming[1,2]. Information appertaining to the pathways that underlie these phenomena is largely fragmentary. Until now, interest has focused mainly on fathoming the intricacies of the neuronal networks that regulate the genesis, the maintenance and the termination of REM sleep. This research has revealed the crucial involvement of several brainstem areas, located prevalently in the pons[3]. During REM sleep, postural motoneurons are inhibited, and muscles are relaxed by the activation of a circuit originating from the sublaterodorsal (SLD)/subcoeruleus (SubC) nucleus of the pons[1,4] The motoneurons that subserve the external eye muscles are no exception to this rule; although tonically inhibited, they are phasically activated[5,6], and provoke the rapid EMs that are characteristic of REM sleep. Although the eyes move in tandem during REM sleep[7], the movements are often unconjugated[5]. Temporally, they coincide with the generation of the PGO waves[6,8]—endogenous signals that in rats might correspond to P-waves—which originate in the SLD/SubC[9]. In terms of direction and velocity, EMs during REM sleep differ from the saccades that are generated during wakefulness[5,10]. The source of the phasic motor commands that underlie the EMs of REM sleep is unknown[11,12].

In this study, we identify a conserved cluster of Calbindin-D28k (Calb)-immunoreactive neurons in the *medulla oblongata*. On the basis of the results gleaned from in vivo neuroanatomical and optogenetic experiments, we conclude that these neurons contribute to the control of EMs during REM sleep.

## Results

**The *Nucleus papilio*: a conserved cluster of Calbindin-expressing neurons.** We have previously identified in the rat brain a symmetric cluster of Calb-expressing neurons, lodged at the dorsomedial boundaries of the dorsal paragigantocellular nucleus (DPGi), distal to the abducens nucleus (6N), lateral to the medial longitudinal fascicle (mlf), and below the *Nucleus prepositus hypoglossi* (Pr)[13]. This butterfly-shaped cluster of Calb-expressing neurons, here referred to as NP$^{Calb}$ (*Nucleus papilio*), is phylogenetically conserved in rodents (Fig. 1a–c), cats (not shown), monkeys (Fig. 1d), and humans (Fig. 1e, f). It spans a distance of ~ 0.6 mm (Bregma: −5.8 to −6.4 mm[14]) in mice, ~1.9 mm (Bregma: −10.8 to −12.7 mm[15]) in rats and ~ 5–8 mm in humans, in which it originates close to the abducens nucleus and extends caudally. Using stereological techniques, we have estimated the cell density in the NP$^{Calb}$ per hemisphere to be 388.1 ± 47.9 neurons in mice, 849.1 ± 122.1 neurons in rats and 1776.1 ± 116.7 neurons in humans.

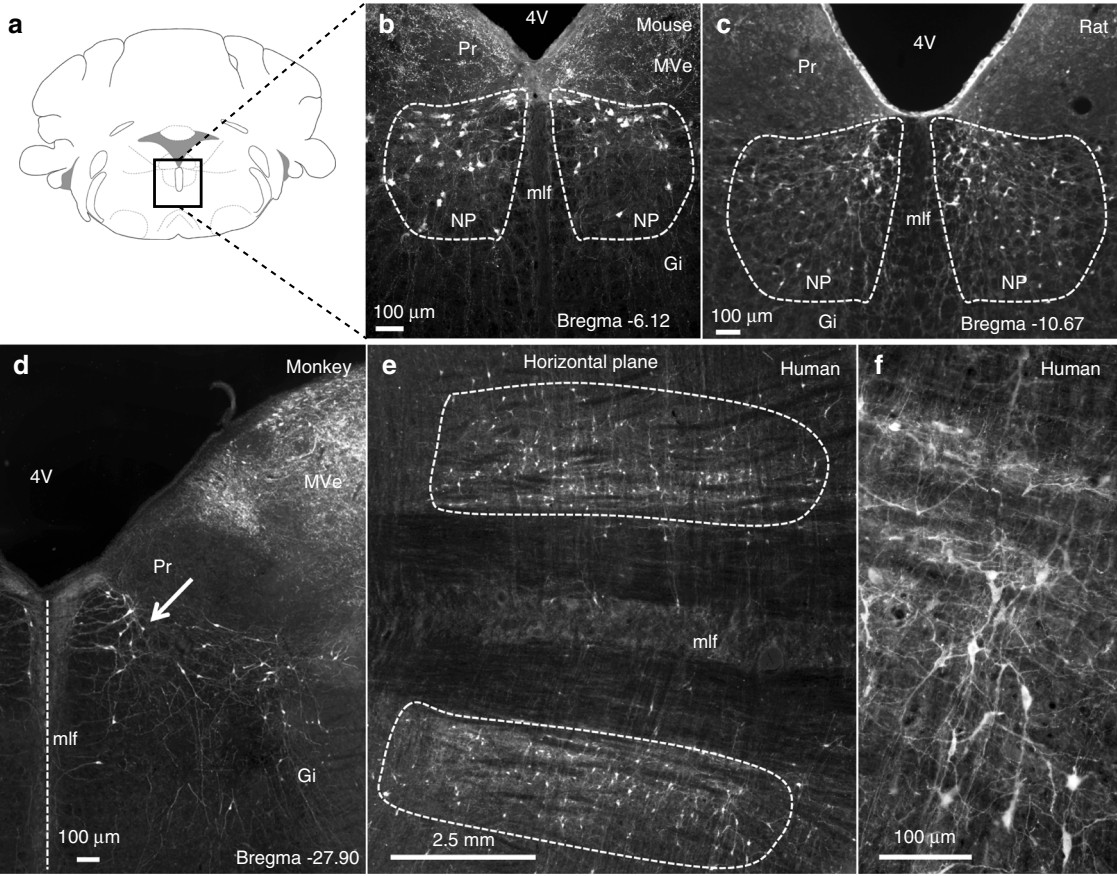

**Fig. 1** The *Nucleus papilio* contains Calbindin-D28k expressing neurons. **a** Schematic representation of a coronal section through a rodent brain. **b–f** Calb-immunoreactivity is observed in neuronal cell bodies and processes in the NP$^{Calb}$ of mice **b**, rat **c**, monkeys **d** and humans **e**, **f**. The NP$^{Calb}$ is delineated with white dashed lines **b**, **c**, **e** or with an arrow **d**. A vertical dashed line in **d** indicates the midline. The human brains **e**, **f** were sectioned in the horizontal plane, the others in the coronal plane. **f** Higher magnification showing dendrites of Calb-immunoreactive neurons of the human NP$^{Calb}$ within the reticular formation. 4V: fourth ventricle; Pr: *N. prepositus hypoglossi*; mlf: medial longitudinal fasciculus; Gi: gigantocellular reticular nucleus; MVe: medial vestibular nucleus; NP: *Nucleus papilio*

**Activity of NP[Calb] neurons during REM sleep.** The DPGi nucleus contains REM-on neurons (viz, those specifically activated during REM sleep)[16], which play a role in the genesis of REM sleep via a GABAergic-mediated inhibition of wake-promoting neurons that are located in the *Locus coeruleus* (LC) and the dorsal raphe nucleus (DR)[17–20]. Although chemical inactivation of these DPGi-neurons induces a prolonged period of wakefulness[17], neurotoxic injury of the DPGi distal to the abducens region suppresses EMs without affecting the total duration of sleep or the bout duration of REM sleep[21]. After instigating a protocol of REM sleep deprivation and selective REM sleep rebound in both mice and rats, a large proportion of the NP[Calb] neurons were found to co-express c-fos, a marker of neuronal activity (the proportion of Calb-immunoreactive neurons expressing c-fos being $46.5 \pm 9.3\%$ in rats and $42.2 \pm 5\%$ in mice; Fig. 2a–c). Furthermore, $53.35 \pm 11.89\%$ ($n = 4$ rats) of the c-fos-labelled neurons in the NP[Calb] failed to express Calb. These results were further supported by the single-cell electrophysiological recording of opto-tagged NP[Calb] neurons in freely moving mice. To achieve this end, we targeted the expression of channelrhodopsin-2 (ChR2) in NP[Calb] neurons by stereotactically injecting into the NP[Calb] of *Calb1*::Cre driver mice the adeno-associated virus encoding ChR2, which was fused to an enhanced yellow fluorescent protein reporter (AAV-EF1a-DIO-hChR2(H134R)-YFP) (Fig. 2d). During REM sleep, the firing activity of the opto-tagged NP[Calb] neurons was augmented relative to that in other cells of the NP[Calb] area (viz., non-responders) (REM: $6.9 \pm 0.5$ Hz in responder cells versus $0.025 \pm 0.0572$ Hz in non-responders; Fig. 2e). Importantly, the firing rate of putative NP[Calb] neurons significantly increased prior to EMs during REM sleep, with a latency of $1.2 \pm 0.8$ s ($n = \sim 40$ EMs registered in 13 recording sessions in $n = 4$ mice; Fig. 2f–h). Together, these data demonstrate that, during REM sleep, the activity of NP[Calb] neurons is time-locked to EMs.

**Anterograde and retrograde tracing of NP[Calb] neurons.** To reveal the neuronal circuit that underlies the control of EMs during REM sleep, the efferences of NP[Calb] neurons were mapped by stereotactic injections of adeno-associated, Cre-dependent viral tracers (AAV2/1.CAG.FLEX.Tomato.WPRE.bGH or AAV-EF1a-DIO-hChR2(H134R)-YFP) into the NP[Calb] of *Calb1*::Cre driver mice. Genetic targeting was specific for the Calb-immunoreactive neurons, stable and confined to the NP[Calb] neurons for at least 5 weeks after the injection (Fig. 3a top). Anterograde mapping (Table 1) revealed a large number of axonal endings in the contralateral nuclei that coordinate the EMs (oculomotor (III), trochlearis (IV) and, to a lesser degree, the abducens (VI)) (Fig. 3b; Supplementary Figs 1d and 2). Axonal terminals were also observed in the Darkschewitsch nucleus, the Roller nucleus (Supplementary Fig. 1b) and the *raphe interpositus*, the latter region composing the omnipause centre that blocks the EMs of the waking state[22]. The SLD/SubC-nuclei (involved in the generation of the rodent analogue of the PGO waves and in the induction of atonia in the postural muscles) and the pontine reticular nucleus (PnC) (involved in the genesis and the maintenance of REM sleep) likewise receive inputs (Supplementary Fig. 1e, g). Terminals were also observed in the areas that are implicated in the processing of sensory inputs (nucleus of the trapezoid body, cuneate- and vestibular nuclei) and in lingual movements (hypoglossal nucleus) (Supplementary Fig. 1b, h). In contrast to the previously reported REM-on GABAergic neurons in the DPGi[18], the NP[Calb] neurons project neither to the LC nor to the DR. The distribution in the eye-muscle nuclei of the axonal terminals from NP[Calb] neurons indicates a targeting of nerve cells supplying the single-innervated, twitch-muscle fibres[23]. We found that $\sim 33.8\%$ of NP[Calb] neurons co-expressed the *Slc17a6* gene

transcripts (encoding the glutamate transporter VGlut2; Supplementary Fig. 3b). Their axonal terminals in the oculomotor nuclei manifested immunoreactivity for the VGlut2 protein (Supplementary Fig. 3c). No co-expression of the GABA transporter VGAT was observed (Supplementary Fig. 3b). Hence, this projection apparently utilises glutamate as a neurotransmitter. Furthermore, trans-synaptic retrograde mapping of NP[Calb] neurons using a Rabies virus (Env-ΔG-Rabies-GFP + AAV-B19G) revealed monosynaptic inputs from brain regions that are known to control REM sleep[24,25], including the lateral hypothalamus (harbouring the REM sleep-promoting, melanin-concentrating-hormone (MCH)-positive neurons), the SubC and the pontine reticular nuclei (PnC/PnO) (Table 2; Supplementary Fig. 4).

**Involvement of NP[Calb] neurons in the control of EMs during REM sleep.** Given the anatomical connections between the NP[Calb] neurons and the three ocular motor nuclei, the functional influence of these neurons on EMs was directly assessed using different but complementary experimental approaches.

Initially, we stereotactically injected the AAV-EF1a-DIO-hChR2(H134R)-YFP virus into the NP[Calb] of *Calb1*::Cre mice. These mice were then chronically implanted with optical fibres, tetrodes and EEG/EMG/EOG electrodes for the simultaneous electrophysiological recording of single-unit activity across sleep–wake states in freely moving mice. The resulting expression of ChR2-eYFP was specific, and confined to the NP[Calb] neurons (Fig. 3a bottom) and their axonal terminals in the 3N (Fig. 4b). Optical stimulation in freely moving mice reliably induced action potentials in ChR2-expressing NP[Calb] neurons during REM sleep and to a much lesser extent during wakefulness (Fig. 3c). Consistent with our correlative data, optogenetic activation of the ChR2-expressing NP[Calb] neurons upon 1 s continuous light, or 10 Hz, stimuli consistently enhanced the probability of evoking EMs during REM sleep within a 1 s window relative to control conditions (10 Hz: $58.6 \pm 5.5\%$ vs $27.3 \pm 3.5\%$; 1 s: $77.2 \pm 3.4\%$ vs $17.2 \pm 3.5\%$; Fig. 3f); so too did optical stimulation with 1 Hz-pulse, albeit to a lesser extent ($56.7 \pm 7.0\%$ vs $32.9 \pm 3.0\%$; Fig. 3f). Average latencies between the optical stimulus and the onset of EMs were significantly shorter for the 1 s-pulses than for the other stimuli in the ChR2-transduced mice relative to those in the control animals (1 s: $0.8 \pm 0.06$ s vs $5.8 \pm 1.7$ s; 10 Hz: $1.8$ s $\pm 0.6$ vs $5.7 \pm 1.9$ s, 1 Hz: $2.8 \pm 0.3$ s vs $7.1 \pm 0.8$ s; Fig. 3g). When the optical stimuli were delivered during the state of wakefulness or during NREM sleep, no changes in the probability of, or latency to, EMs were observed in the ChR2-transduced mice relative to the control animals (Supplementary Fig. 5).

To ascertain whether a direct activation in the 3N of the axonal terminals from NP[Calb] neurons support EMs during REM sleep, the NP[Calb] in *Calb1*::Cre mice was stereotactically injected with AAV-EF1a-DIO-hChR2(H134R)-YFP. The animals were chronically implanted with a single optical fibre, which was positioned just above the 3N (Fig. 4a). Optogenetic activation of the CHR2-expressing NP[Calb] neuronal terminals in the 3N (Fig. 4b) significantly enhanced the probability of evoking EMs relative to control conditions during REM sleep ($76.17 \pm 3.6\%$ vs $30.17 \pm 2.83\%$ Fig. 4e) but not during wakefulness ($44.94 \pm 4.66\%$ vs $38.62 \pm 3.26\%$ Fig. 4e). These data are consistent with those gleaned from the optogenetic activation of the cell body. The latency to EMs was significantly shorter in this experiment (ChR2 vs Ctrl: $0.63 \pm 0.02$ s vs $8.88 \pm 1.26$ s; Fig. 4f), than in that targeting the activation of the cell body of NP[Calb] cells (ChR2 *vs* Ctrl: 1 s: $0.8 \pm 0.06$ s vs $5.8 \pm 1.7$ s; Fig. 3g). Importantly, optogenetic activation of either the cell body or the axonal terminals of NP[Calb] neurons had no significant effect on either the duration (Fig. 4g, h, respectively) or the frequency of REM sleep episodes

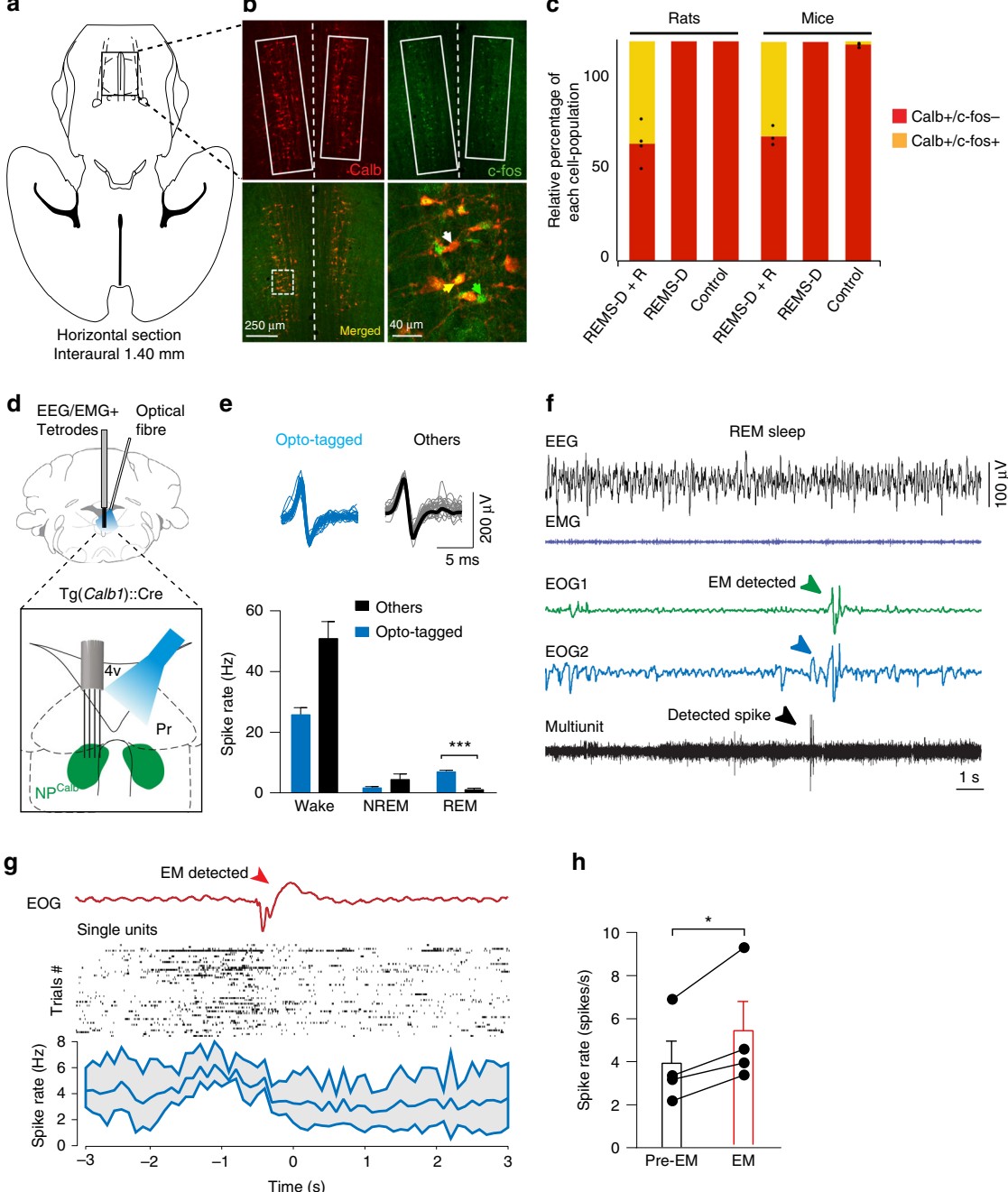

**Fig. 2** Calbindin-D28k-expressing neurons of the *Nucleus papilio* are active during REM sleep. **a** Schematic representation of the localisation of the NP^Calb in a rat horizontal brain section. **b** After a period of REM sleep deprivation, followed by a 3 h REM sleep rebound ("REMS-D+R"), a large subset of NP^Calb neurons (boxed in white) expressed c-fos protein. Shown are representative examples of immunostaining in rat brain. A higher magnification view of the delineated area reveals the three populations of cells, namely, the neurons that are immunoreactive only for Calb (white arrow), those that are immunoreactive only for c-fos (green arrow), and those that are immunoreactive for both markers (yellow arrow). **c** Relative percentage of cell immunoreactive for c-fos in "REMS-D+R" ($n = 3$ mice and $n = 4$ rats), "REMS-D" ($n = 3$ mice and $n = 2$ rats) and "Control" ($n = 3$ mice and $n = 3$ rats) groups. In the REMS-D+R group almost half of the Calb-immunoreactive neurons ($42.2 \pm 5\%$ in mice, and $46.5 \pm 9.3\%$ in rats) were c-fos-positive. **d** Schematic representation of the movable tetrodes and the optical fibre implanted in the NP^Calb. **e** NP^Calb neurons were identified by their spiking response to single light pulses. Top: averaged neuronal spike rates of opto-tagged cells (blue) and non-responders (black) were similar. Bottom: averaged neuronal spike rates of opto-tagged cells (blue) and non-responders (black) across sleep–wake states. NP^Calb neurons display significant lower firing rate than non-responders during wake and the two populations have the opposite behaviour during REM sleep ($n = 4$ responder cells, and $n = 6$ non-responder cells; four animals; wake: $P = 0.0086$, $t = 3.45$; NREM: $P = 0.289$, $t = 1.35$ and REM: $P \leq 0.0001$, $t = 7.29$; $df = 8$; un-paired two-tailed $t$ test). **f** EEG/EMG representative trace during REM sleep, electro-oculogram (EOG) signals and detected EM (green/blue arrows) showing the correspondent multiunit activity and a representative detected spike (black arrow). **g** Top traces represent EOG, unfiltered signal. The raster plot of the spiking activity of opto-tagged NP^Calb neurons (black) and the corresponding averaged spiking rates (blue trace) reveal the increase in firing activity that preceded the EMs during REM sleep. **h** Average summary data during REM sleep of the spiking rate of single-cell activity before and during EM (pre-EM = firing before EM: $3.901 \pm 1.033$ spikes/s; EM = firing at EM: $5.299 \pm 1.353$ spikes/s; $n = 4$ cells; $P = 0.0277$; $t = 4.020$; $df = 3$: two-tailed paired $t$ test). Error bars represent SEM values

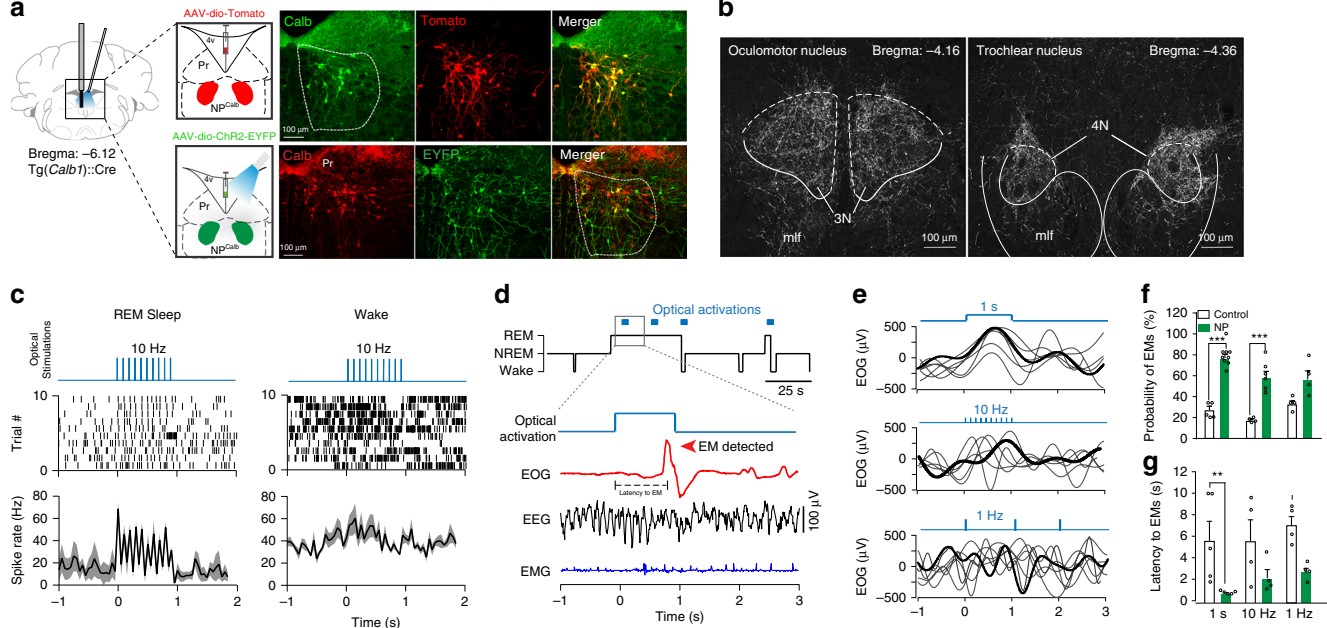

**Fig. 3** Optogenetic activation of NP$^{Calb}$ neurons induces EMs during REM sleep. **a** Schematic representations of the stereotactic virus injections in the brain of *Calb1*::Cre mice (top: tracing experiment with Cre-dependent AAV-Tomato (AAV2/1.CAG.Flex.Tomato.WPRE.bGH); Bottom: optogenetics experiment with AAV-EF1a-DIO-hChR2(H134R)-YFP), tetrodes and optical fibre placement above the NP$^{Calb}$ (EEG/EMG/EOG electrodes are not shown). For each of the two experimental paradigms, representative images of coronal sections show expression of either Tomato and Calb (upper row) or ChR2-YFP and Calb (lower raw) in NP$^{Calb}$ neurons. The NP$^{Calb}$ region is delineated with white dashed lines. 4V: fourth ventricle; Pr: *N. prepositus hypoglossi*. **b** Dense Tomato-expressing axon terminals of NP$^{Calb}$ neurons were found in the oculomotor (3N) and trochlear (4N) nuclei. The site of injection was median, resulting in the infection of NP$^{Calb}$ neurons in both hemispheres. mlf: medial longitudinal fasciculus. **c** Light-evoked spiking response of opto-tagged NP$^{Calb}$ cells during REM sleep (left) and wakefulness (right). Raster plots show the spiking activity of a representative NP$^{Calb}$ neuron during each state. Optical stimuli are shown on the top scheme. Bottom: averaged neuronal spike rates ± SEM ($n = 4$ cells in four mice) during REM sleep and wakefulness. **d** Representative hypnogram of optogenetic activation of NP$^{Calb}$ neurons in vivo (top) together with EEG, EMG and EOG recordings. Red arrow indicates a light-evoked EM. The latency to EM is also indicated with a dashed line. **e** Overlay of averaged EOG recordings showing EMs triggered by optogenetic stimulation of NP$^{Calb}$ neurons at different frequencies during REM sleep. Grouped data and single animal values are shown in black and grey, respectively. **f**, **g** Averaged probabilities **f** and latencies **g** of EMs in responses to optogenetic stimulations of NP$^{Calb}$ neurons during REM sleep. The data derived from at least five REM sleep episodes per stimulation frequency in ChR2-transduced and control animals (probability of EM: 1 s continuous light ($n = 9$ and 5, respectively; $P = $ <0.001, $t = 8.818$, $df = 26$), 10 Hz ($n = 4$ and 6, $P = $ <0.001, $t = 6.147$, $df = 26$) and 1 Hz ($n = 4$ and 4; $P = 0.014$, $t = 3.228$, $df = 26$). Latency to EM: 1 s continuous light ($n = 9$ and 5, respectively; $P = 0.036$, $t = 3.765$, $df = 20$), 10 Hz ($n = 4$ and 4; $P = 0.1103$, $t = 2.219$, $df = 20$) and 1 Hz ($n = 4$ and 4; $P = 0.224$, $t = 1.787$, $df = 20$)). $P = 0.12$ ns (*), 0.33 (*), 0.002 (**), < 0.001 (***), two-way ANOVA followed by Sidak post hoc test for multiple comparisons. Error bars represent SEM values

(Supplementary Fig. 6), or on the parameters of either NREM sleep or wakefulness (Supplementary Fig. 6).

Finally, we adopted two different lack-of-function approaches to ascertain whether NP$^{Calb}$ neurons were necessary to generate the EMs during REM sleep. Initially, an adeno-associated virus encoding a Cre-dependent diphtheria toxin receptor (DTR) (AAV9-pCAG-Flex-DTR) was stereotactically delivered to the NP$^{Calb}$ of *Calb1*::Cre mice. The animals were then chronically implanted with EEG/EMG/EOG electrodes (Fig. 5a). After habituation to intraperitoneal (i.p.) injections of saline for 3 days, the diphtheria toxin (DTX) was administered i.p. on day 0, and EEG/EMG/EOG signals were continuously recorded for three successive days (Fig. 5b), after which period most of the NP$^{Calb}$ neurons were ablated (>93%, Fig. 5c, d). During REM sleep, the number of EMs decreased markedly in the DTR-transduced mice relative to that in the control animals ($2.1 ± 0.8$ vs $16.0 ± 1.1$ on day 3; Fig. 5e, f), whereas they remained unchanged during wakefulness ($32.57 ± 3.57$ vs $40.0 ± 1.78$ on day 3; Fig. 5g). No statistically significant changes in sleep patterns were observed after DTX treatment in the DTR-transduced and the control animals (Supplementary Fig. 7). To substantiate our findings, the AAVdj-EF1alpha-DIO-ArchT3.0-EYFP virus encoding the archeorhodopsin 3.0 proton pump was stereotactically injected into the

NP$^{Calb}$ of *Calb1*::Cre mice to induce the transient optogenetic silencing of NP$^{Calb}$ neurons selectively during REM sleep in freely moving mice (Fig. 5h, i). Consistent with the genetic ablation experiment, optogenetic silencing of the NP$^{Calb}$ neurons induced a significant reduction in the total number of EMs that were evoked during a single episode of REM sleep in the ArchT3.0-transduced mice relative to that in the control animals ($4.4 ± 1.5$ vs $18 ± 1.1$; Fig. 5k). The duration of an episode of REM sleep (Fig. 5l), and other sleep features (Supplementary Fig. 7), remained unaffected. Collectively, these results confirm the necessity of NP$^{Calb}$ neurons in the control of EMs, specifically during REM sleep.

## Discussion

The data gleaned from investigations using different approaches, namely transectioning, lesioning or chemical inhibition, have established the pons to be the region of the brain responsible for the evocation of REM sleep, EMs, PGO-spikes and atonia[1,8,26–31]. For example, lesioning of the pedunculo-pontine tegmentum lead to a marked reduction in the number of EMs and of PGO-spikes[30]. However, some of these pontine lesions may have intersected the pons-traversing pathway from the NP$^{Calb}$ in the upper medulla to the nuclei of the external eye muscles in the midbrain.

**Table 1 Main efferent connections**

**Brain regions/identified nuclei—principal functions and pathways**

*Medulla*
CeCv, central cervical nucleus
MdD, medullary reticular nucleus, dorsal
MdV, medullary reticular nucleus, ventral
12N, hypoglossal nucleus—innervation of lingual muscles; regulated during sleep
Ro, nucleus of roller—part of circuits related to eye movements/sends imputs to III, IV and VI
LRt, lateral reticular nucleus
LPGi, lateral paragigantocellular nucleus—REM-on; proposed as inhibiting PS-off neurons of the LC; control of penile erection
Gi, gigantocellular reticular nucleus—inhibition of motor neuron activity in the spinal cord during sleep
Spve/Mve/Lve/SuVe, spinal/medial/lateral/superior vestibular nucleus—coordination of head and eye movements
MVePC/MVeMC, medial vestibular nucleus parvicellular/magnocellular—coordination of head and eye movements
6N, abducens nucleus (and Pa6, para-abducens nucleus)—oculomotor system (coordination of eye movements in the same direction)
*Pons*
Tz, nucleus of the trapezoid body—part of the auditory pathways
LSO, lateral superior olive—part of the auditory pathways
DPO, dorsal periolivary region—part of the auditory pathways
Bar, Barrington nucleus—principal micturition centre
RIP, raphe interpositus nucleus—premotor network for saccades (omnipause neurons)
SubC, subcoeruleus nucleus—induces muscle atonia during REM sleep; source of the PGO waves (rodents)
IRt, intermediate reticular nucleus
KF, Kölliker-Fuse nucleus—modulation of the respiratory function
PnC, pontine reticular nucleus, caudal part—REM sleep initiation and maintenance; contains premotor neurons for saccades
*Midbrain*
4N, trochlear nucleus (and Pa4, paratrochlear nucleus)—oculomotor system (innervate the superior oblique muscle of the contralateral eye)
3N, oculomotor nucleus (and 3PC, oculomotor nucleus parvicellular)—oculomotor system (innervate external eye muscles except superior oblique and lateral rectus)
isRT/mRT, isthmic/mesencephalic reticular formation—involved in oculomotor function
Dk, nucleus of Darkschewitsch—one of the accessory oculomotor nuclei (eye movements, gaze coordination)

Summary of the main efferent connections of the *Nucleus papilio* identified by anterograde-tracing experiments with Cre-dependent AAV tracers in *Calb1*::Cre mice ($n = 5$ mice). The principal functions and pathways in which these nuclei are involved is indicated, with a special emphasis on sleep and eye movement regulation

**Table 2 Main afferent connections**

**Brain regions/identified nuclei**

*Medulla*
MdD/MdV, medullary reticular nucleus dorsal/ventral
Sp5C/Sp5I, spinal trigeminal nucleus caudal/interpolar
LPGi, lateral paragigantocellular reticular nucleus
Gi, gigantocellular reticular nucleus
DPGi, dorsal paragigantocellular nucleus
Spve/Mve/Lve/SuVe, spinal/medial/lateral/superior vestibular nucleus
MVePC/MVeMC, medial vestibular nucleus parvicellular/magnocellular
Irt, intermediate reticular nucleus
*Pons*
SubC, subcoeruleus nucleus
PnC, pontine reticular nucleus, caudal part
PnO, pontine reticular nucleus, oral part
*Midbrain*
mRT, mesencephalic reticular nucleus
*Hypothalamus*
rare sparse neurons in the hypothalamus

Summary of the main afferent connections to the *Nucleus papilio* identified by retrograde-tracing experiments with rabies virus in *Calb1*::Cre mice ($n = 5$ mice)

The DPGi, which was the focus of the present investigation, appears to be situated at the most caudal limit of this critical pontine region. The DGPi harbours two radically different types of neurons: the glutamatergic NP^Calb neurons described here, and the GABAergic ones[16–20]. Although the latter are involved in the inhibition of waking neurons, particularly those from the LC, the NP^Calb neurons project excitatory connections to the nuclei of the external eye muscles.

The anatomical localisation of the NP^Calb in the dorsomedial DPGi, together with its glutamatergic innervation of each of the three contralateral oculomotor nuclei that are responsible for the generation of EMs, distinguish these cells from the saccade-generating premotor circuit that is responsible for the generation of EMs during wakefulness. The vertical and the torsional directions of the EMs during wakefulness are co-ordinated by the rostral interstitial nucleus of the medial longitudinal fascicle in the midbrain[32], whereas those in the horizontal direction are under the control of excitatory, glutamatergic (EBNs) and inhibitory, glycinergic burst neurons (IBNs), which are respectively located rostral (EBNs) and just caudal (IBNs) to the 6N[33,34]. Unlike the NP^Calb, which is comprised of Calb/VGlut2-expressing neurons that do not co-express parvalbumin (Supplementary Fig. 3d), the EBNs and the IBNs do express this calcium-binding protein and are mantled with perineuronal nets[22,35]. Collectively, these differences afford evidence that EMs during REM sleep and wakefulness are, at least partly, controlled by distinct neuronal circuits. Nevertheless, the EOG is a rough readout of EMs, and our finding that the genetic ablation of NP^Calb neurons does not significantly decrease the number of EMs that are evoked during wakefulness does not exclude the possibility that a more refined technique for measuring EMs might reveal subtle differences.

We report here that the NP^Calb is conserved in rodents and primates, including humans, with respect both to its Calb-immunoreactivity and its anatomical position. Several imaging studies in humans (fMRI, tomography, PET) have disclosed the EMs that are evoked during the phases of wakefulness and REM sleep to share several neural correlates, including neuronal activation in the pons and the cortical frontal eye fields[36,37]. Neuronal activity in the *medulla oblongata* of humans has not hitherto been reported in association with EMs during REM sleep, perhaps owing to the technical difficulties of imaging this region of the brainstem.

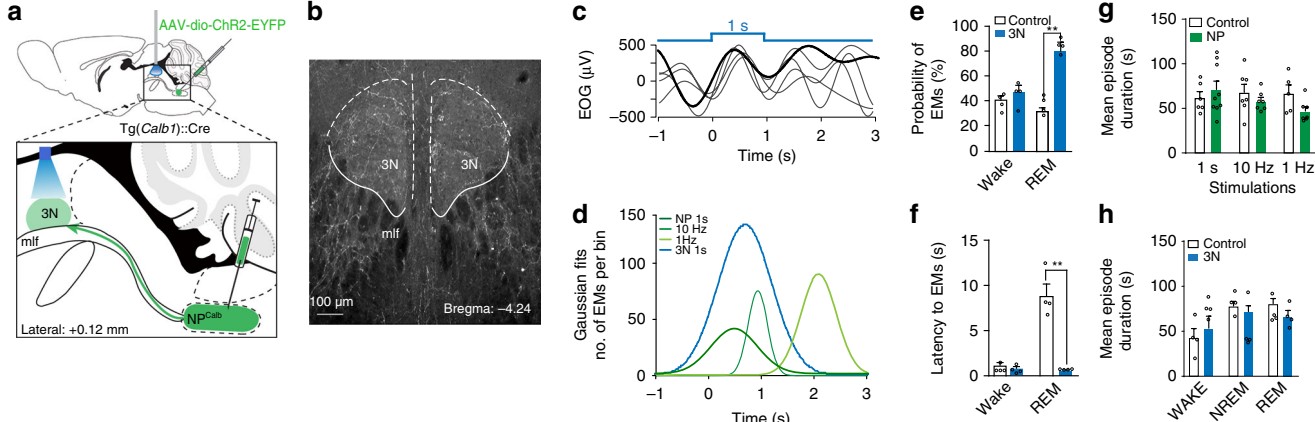

**Fig. 4** Optogenetic activation of NP$^{Calb}$ terminals in the oculomotor nucleus. **a** Schematic representation of the experimental set up used to activate the axon terminals of NP$^{Calb}$ neurons in the oculomotor nucleus (3N). The boxed area represents a higher magnification view of the injection of AAV-EF1a-DIO-hChR2(H134R)-YFP in the NP$^{Calb}$ and the location of the optical fibre used for light delivery above the 3N. **b** Representative image of ChR2-EYFP-positive terminals in the 3N. **c** Overlay of averaged EOG recordings showing EMs triggered by optogenetic stimulation of NP$^{Calb}$ neuron terminals in the 3N upon 1 s continuous illumination during REM sleep. Grouped data and single animal values are shown in black and grey, respectively. **d** Comparative Gaussian distribution of EOG signals triggered by optogenetic activation of NP$^{Calb}$ neuron soma ($n = 6$) or 3N terminals ($n = 4$) per condition. Gaussian distribution is a fit to maximal deflection of EOG signals. **e, f** Averaged probabilities **e** and latencies **f** of EMs in response to optogenetic stimulations of NP$^{Calb}$ neuron terminals in 3N during wakefulness and REM sleep ($n = 4$ mice per group; $P = 0.43$, $t = 0.8895$ for wake; $P = 0.0016$, $t = 11.098$ for REM; $df = 3$ in both experimental conditions tested). *$P < 0.001$ significance was calculated using a paired $t$ test. **g** Summary of the mean duration of wake, NREM and REM sleep bouts, during a 1 h period of light activation delivered every minute of NP$^{Calb}$ cell soma at 1 s continous light ($n = 9$ and 6, $P = 0.79$, $t = 0.832$), 10 Hz ($n = 7$ and 7, $P = 0.69$, $t = 0.876$), and 1 Hz ($n = 7$ and 5, $P = 0.28$, $t = 0.0.65$); $df = 35$, when comparing ChR2-transduced to control animals. **h** Summary of the mean duration of wake, NREM and REM sleep bouts during a 1 h period of 1 s continuous stimulation delivered every minute to activate the NP$^{Calb}$ neuron terminals in 3N ($n = 4$ in each group; $P = 0.78$, $t = 0.8691$ for wake; $P = 0.941$, $t = 0.517$ for NREM; and $P = 0.592$, $t = 1.167$ for REM; $df = 18$ when comparing ChR2-transduced to control animals). Neither the activation of the NP$^{Calb}$ cell soma nor the activation of their terminals in the 3N had a significant effect on the duration of wakefulness, NREM and REM sleep episodes. Error bars represent SEM values

The findings of our optogenetic experiments reveal the firing rate of NP$^{Calb}$ neurons to increase prior to the generation of EMs, thus enhancing the probability of their evocation during REM sleep. The occurrence of a delay between the optogenetic activation of the NP$^{Calb}$ neurons and the manifestation of EMs is reminiscent of a similar delay that is observed between the stimulation of the SubC and the generation of PGO waves, which coincides temporally with the manifestation of EMs[38].

Although no significant effects on the onset and the maintenance of REM sleep parameters were observed either upon the activation, the inhibition or the deletion of NPCalb neurons, further studies are required to ascertain whether the competence of these neurons—or of others intermingling Calb-negative ones in this region—transcend in scope its premotoric role in the generation of EMs during REM sleep.

In summary, our data furnish experimental evidence in a mouse model that NP$^{Calb}$ neurons and their projections to the oculomotor nuclei contribute to the control of EMs during REM sleep. The capability to induce EMs during REM sleep on command affords a powerful tool for the investigation of their functions.

## Methods

**Animals**. For the morphological and the immunofluorescence analysis of the NP$^{Calb}$ in rodents, 8–12 weeks old C57BL/6 J mice (from our animal facility) of both genders ($n = 8$) were utilised, as well 3 months old Wistar rats (Janvier, Lyon, France) ($n = 3$ males and three females).

The REM sleep deprivation experiments were conducted using nine female Wistar rats (Janvier, Lyon, France), and nine C57BL/6 J mice.

The B6.Cg-*Calb1*$^{tm1.1(folA/Cre)/Hze}$/J murine line (*Calb1*::Cre) was purchased from the Jackson Laboratory, and bred in our animal facility under normal conditions. Genotyping was performed by PCR following the protocol of the supplier. For the anterograde-tracing experiments, five *Calb1*::Cre male mice were used. The injections were deemed to have been successful if the tracer was confined to the NP$^{Calb}$ neurons. For the retrograde-tracing experiments, five *Calb1*::Cre mice were employed. For analysing connections of the hypothalamic MCH neurons to

the NP$^{Calb}$, two *Mch*::Cre mice (Tg(*Pmch*-Cre)1Lowl, Jackson Laboratories, Bar Harbour, USA) were employed.

For studying the coexistence of Calb-expression with various markers for neurotransmitters, four mice each of three mouse lines obtained from Jackson laboratory (*Slc17a7*$^{tm1.1(Cre)/Hze}$/J for VGlut1; *Slc17a6*$^{tm2(Cre) lowl}$/J for VGlut2, and *Slc32a1*$^{tm2(Cre) lowl}$/J for VGat) were investigated.

The optogenetic stimulation,—or inhibition—experiments were conducted with 13 *Calb1*::Cre male mice, and 9 control wild-type littermates. The full analysis was retrospectively limited to mice in which the injection of the AAV-ChR2 [AAV2-EF1a-DIO-hChR2(H134R)-YFP] or AAV-ArchT [AAV2-Ef1a-DIO-ArchT3.0-EYFP] viruses and its expression were limited to the NP$^{Calb}$.

The experiments consisting in eliminating NP$^{Calb}$ neurons by means of diphtheria toxin (AAV9-pCAG-Flex-DTR) injection were conducted on seven *Calb1*::Cre mice and four control littermates, injected in the NP$^{Calb}$; the full analysis was limited to mice in which the injection of the virus was most precise.

For in vivo recording of NP$^{Calb}$ neuron activity, four *Calb1*::Cre mice were used and for the optogenetic activation of the axonal endings of NP$^{Calb}$ neurons in the oculomotor nucleus, four *Calb1*::Cre male mice were employed.

The animals were housed in state-of-the-art animal facilities and in accordance with the relevant Swiss laws. The study was approved by the Committee for Animal Experimentation of the Cantons of Fribourg (2015-FR18), respectively, Bern (2015-FR18+) (Switzerland).

For anatomical studies, the animals were anaesthetised with pentobarbital (100 mg/kg of body weight). In a state of narcosis they were perfused via the left ventricle, first with chilled (4 °C) physiological (0.9%) saline and then with chilled (4 °C) 4% paraformaldehyde. The brains were excised and post-fixed overnight in the same fixative. They were then immersed in 0.1 M Tris-buffer (pH 7.3) containing 20% sucrose in preparation for cryo-sectioning.

Specimens of the brains of *Macaca mulatta* and *Macaca fascicularis*, which had been killed for other purposes, were provided by Dr. Grazyna Wieczorek (Novartis, Basel, Switzerland).

Samples of human brains were supplied by Professor Luis Filgueira (Anatomy Unit, University of Fribourg, Switzerland). Specimen 2–15 and 7–15 were derived from male subjects, 70 and 80 years of age who had donated their body to anatomy for teaching and research purposes. They had manifested no signs of neurological disorders.

**Immunofluorescence**. Depending on the nature of the experiment, the various brain specimens were cryosectioned into 30-, 40-, or 80-µm-thick sections (coronal or horizontal plane), which were collected directly in 0.1 M Tris-buffer containing

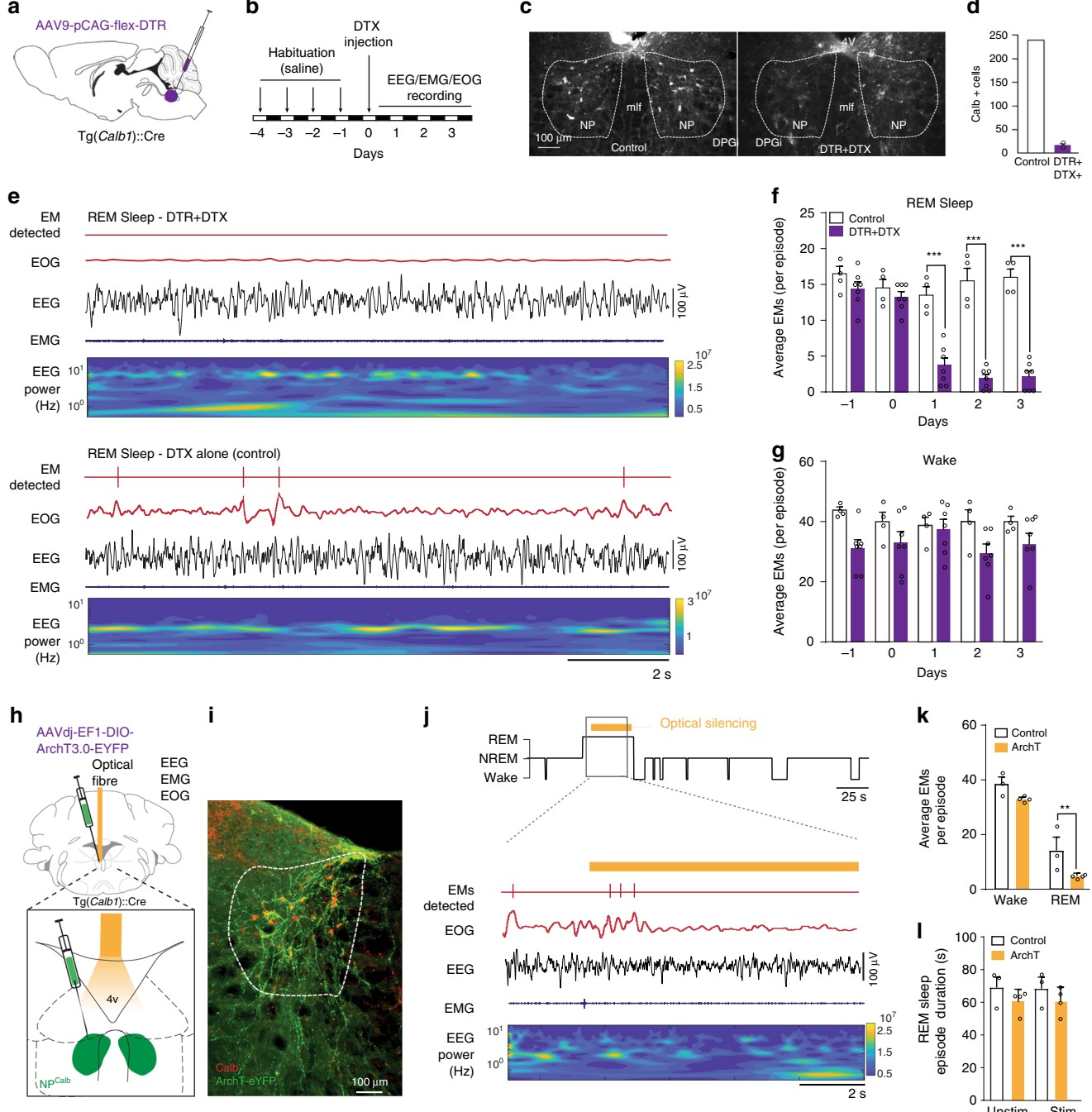

**Fig. 5** Genetic ablation and optogenetic silencing of NP^Calb neurons block EMs during REM sleep but not wakefulness. **a** Schematic representation of the genetic targeting of the diphtheria toxin receptor (DTR) expression to NP^Calb neurons and implantation of optical fibre. **b** Experimental timeline. **c** Representative micrographs depicting immunoreactivity for Calb protein in a non-transduced (left) and a DTR-transduced (right) mouse, both i.p.-injected with diphtheria toxin (DTX). Note the remnants of disintegrating neurons in the extracellular space displaying some immunoreactivity for Calb (right). 4V: fourth ventricle; mlf: medial longitudinal fasciculus; NP: *Nucleus papilio*. **d** The number of Calb-expressing neurons is dramatically reduced upon DTX injection in DTR-transduced ($n = 3$) as compared with a wild-type animal that received DTX alone. **e** Representative experiment showing detected EMs, EOG, EEG and EMG signals, and time frequency heat-map of EEG spectral power from animals transduced with DTR and injected with DTX (upper set of traces) and control mice injected with DTX alone (lower set of traces) during REM sleep. **f, g** Averaged number of EMs during REM sleep **f** and wakefulness **g**, before and after DTX injection ($n = 7$ DTR+/DTX+ and four control animals; Wakefulness: $P = 0.523$, F (4, 36) = 0.8171; REM sleep: $P \leq 0.001$, f (4,36) = 19.2). ***$P < 0.0001$, two-way ANOVA followed by Sidak's post hoc tests). Genetic ablation of NP^Calb neurons does not significantly decrease the number of EMs during wakefulness but eradicate EMs during REM sleep. **h** Schematic representation of the genetic targeting of AAV2-Ef1a-DIO-ArchT3.0-EYFP inhibitory opsin expression to NP^Calb neurons and implantation of optical fibre. **i** Representative micrograph showing ArchT-YFP expression (green) in Calb-immunoreactive neurons (red). **j** Experimental strategy for optical silencing of NP^Calb neurons selectively during REM sleep (top). Detected EMs, EOG, EEG and EMG traces and time frequency heat-map of EEG spectral power (bottom). Orange bar represents the duration of the optical silencing. **k** Averaged number of EMs during wakefulness and REM sleep in the control ($n = 3$) and ArchT3.0 ($n = 4$) transduced mice. **, $P = 0.11$, two-way-ANOVA followed by Sidak's multiple comparison test. Error bars represent SEM values. **l** Duration of REM sleep episodes under conditions in which pulses of light were either not delivered (unstim.) or delivered (stim.) to silence the NP^Calb neurons, in control and transduced mice. Error bars represent SEM values

0.02% sodium-azide, and within which they were maintained until the time of analysis. The sections were immuno-stained according to standard protocols[39]. The following primary antibodies were used in this study: mouse and rabbit anti-Calbindin-D28k (CB300 respectively CB38; Dilution: 1:2000; Swant, Marly, Switzerland), mouse and rabbit anti-c-fos (ab208942 respectively PC38, Dilution: 1:2000; respectively, Abcam, Cambridge, UK and Merck, Darmstadt, Germany), rabbit anti-GFP (A11122; Dilution: 1:2000; Life Technologies Europe, Zug, Switzerland), rabbit anti-MCH (H-070-47; Dilution: 1:1000; Phoenix Pharmaceuticals, Karlsruhe, Germany), mouse and rabbit anti-parvalbumin (PV235, respectively, PV28; Dilution: 1:1000; Swant, Marly, Switzerland), mouse anti-tyrosine hydroxylase (22941; Dilution: 1:1000; Immunostar, Houston, USA), rabbit anti-VGlut2 (135403; Dilution: 1:2000; Synaptic Systems, Goettingen, Germany). The secondary antibodies included Cy3-conjugated anti-rabbit and anti-mouse, Cy3- and Cy2-conjugated streptavidin, Cy2-conjugated anti-mouse, Alexa488-conjugated anti-rabbit and anti-mouse (Jackson Immunoresearch, Suffolk, UK), biotinylated anti-rabbit and anti-mouse (Vector Laboratories, Servion, Switzerland).

The number of Calb-immunoreactive neurons in the brains of various species ($n = 6$ mice; $n = 6$ rats; $n = 2$ humans) was quantified using Stereoinvestigator 10.52 software (MBF Bioscience, Williston, USA) mounted on a Zeiss photomicroscope, which was equipped with a Hamamatsu Orca OG5 camera.

**Deprivation of REM sleep and rebound.** Rats (female Wistar rats, Janvier, Lyon, France) and mice (from our animal facility) were deprived of REM sleep by implementing a modified version of the flower-pot technique, which spared the animals of major stress[40,41]. Three groups were established: in the first group ("REM sleep deprivation and rebound" = REMS-D+R), the animals ($n = 4$ rats; $n = 3$ mice) were maintained together for 72 h on six small stone platforms ($7 \times 4$ cm for rats, $3 \times 3$ cm for mice), placed in a water tank. The surface of the platform was 1 cm above the water level. During this 72 h period, the animals had free access to food and water. Owing to the loss of the muscular tone that characterises the onset of REM sleep, the animals fell into the water and were thereby deprived of REM sleep. After 72 h, the animals were transferred to a conventional cage in a quiet room and were permitted for 3 h to undergo REM sleep (= rebound). In the second group ("REM sleep deprivation" = REMS-D), the animals ($n = 2$ rats; $n = 3$ mice) were killed immediately after the termination of the 72 h REM sleep deprivation period, without recovery. In the third group ("control"), the animals ($n = 3$ rats; $n = 3$ mice) were maintained in their cages under standard conditions for 72 h prior to killed. The animals were anesthetized and perfused with fixative as described above under "Animals" subheading. The brains were then excised and cryosectioned. The sections were immuno-stained for Calb as well as for c-fos, a surrogate marker of neuronal activity[42]. The area within which the cells were counted (on horizontal sections for rats and coronal sections for mouse) was delimited by the boundaries of immunoreactivity for Calb. Within this area the number of cells that manifested immunoreactivity for either Calb alone, c-fos alone, or both Calb and c-fos, were estimated by a single observer in a blinded manner with respect to the treatment conditions. For rats, three alternating horizontal sections through each hemisphere (to avoid counting the same cells on the two sides) were analysed. For mice, the analysis involved 10 alternating coronal sections.

**Tract-tracing experiments.** Anterograde-tracing experiments were performed on Calb1::Cre mice weighing 25–30 g, following standard protocol[39]. Three successfully injected mice (those in which labelling was confined to the NP$^{Calb}$ neurons) were used for the detailed analysis. One of the two different Cre-dependent viral constructs, namely AAV2/1.CAG.Flex.Tomato.WPRE.bGH or AAV2/1.CAG.Flex.eGFP.WPRE.bGH (Vector Core, University of Pennsylvania, USA), were stereotactically injected in the NP$^{Calb}$ of Calb1::Cre mice at the following Bregma coordinates: rostro-caudal: −6.36 mm; medio-lateral: −0.2 mm and dorso-ventral: −4.35 mm[14]. Comparable and reproducible results were obtained with the two viral constructs. Four weeks after the stereotactic injections, the animals were anaesthetised and perfused with paraformaldehyde (PFA) 4%. The brains were excised and cryosectioned and the specimens were analysed by immunofluorescence.

Retrograde tracing was performed using an attenuated rabies virus technique[43,44]. Initially, the NP$^{Calb}$ of Calb1::Cre mice ($n = 5$) was stereotactically injected (at the aforementioned coordinates) with a Cre-dependent helper AAV-TVA-Cherry virus. After a lapse of 3 weeks, which sufficed for the expression of the TVA rabies receptor, the following constructs were co-injected: an EnvA-ΔG-Rabies-GFP virus, a Cre-dependent helper AAV-B19G virus (expressing the B19G-protein that is necessary for the spreading of the rabies virus to synaptically connected neurons) and (for only two of the five mice) an anterograde tracer (AAV2/1.CAG.Flex.Tomato.WPRE.bGH). Ten days later, the animals were anesthetized and perfused with 4% PFA. The brains were excised and cryosectioned and the specimens analysed by immunofluorescence.

**Optogenetic experimental procedures and in vivo neuronal recording.** The Cre-dependent channelrhodopsin viral construct [AAV-EF1a-DIO-hChR2 (H134R)-YFP] was stereotactically injected into the NP$^{Calb}$ of Calb1::Cre mice at the aforementioned coordinates. After 4–6 weeks after injections, animals were

chronically implanted with a 200-μm fibre implant (MFC-200/245-0.37-5-TF2-FLT; Doric Lenses) above the NP$^{Calb}$ (AP: −6.36 mm; ML: 0 mm; DV: −4.2 mm) or above the 3N (AP: −4.0 mm; ML: 0 mm; DV: 3.0 mm), and affixed to the skull with C&B Metabond (Patterson dental). Then, custom-made EEG/EMG/EOG implants were placed and secured to the skull with dental cement (Patterson dental). EEG signals were recorded from 4 electrodes on the frontal (AP, −2 mm; ML, ±2.5 mm) and temporal (AP, 3.5 mm; ML, ±3 mm) cortices. EMG signals were recorded from two electrodes inserted in the neck musculature to record postural tone. EOG signals were recorded from bilaterally implanted stainless steel electrodes with a golden tip wire placed into the orbital cavity to record eye movements. After surgical procedures, mice were allowed to recover in an individual housing cage for at least 2 weeks. After 1 additional week of acclimation to the EEG–EMG recording set up, an optical patch cord (MFP-200/230/900-0.37-2m-FC-ZF1.25) and a zirconia sleeve (ID = 1.25; Doric Lenses) were connected permanently on the fibre implant. Black nail polish was applied on the plug to blackout the light during the optogenetic stimulation.

To demonstrate the successful optogenetic stimulation of NP$^{Calb}$ neurons during in vivo experiments, we injected Calb1::Cre animals with AAV-EF1a-DIO-hChR2(H134R)-YFP targeted to the NP$^{Calb}$ (AP: −6.36 mm, ML: 0 mm; DV: −4.2 mm) and then chronically implanted fibre optics and four tetrodes attached to a Microdrive (Nanodrive, Cambridge Neurotech) as well as EEG/EMG four weeks later. Tetrodes were made from four strands of 10-μm twisted tungsten wire (CFW0010954, California Fine Wire) connected to an electrode interface board by gold pins (EIB 36, Neuralynx). For data collection mice were connected to a tethered digitising headstage (RHD2132, Intan Technologies) and data sampled at 20 kHz recorded in free open-source software (RHD2000 evaluation software, Intan Technologies). Optical fibres were connected to a patch chord using a zirconia sleeve (Doric Lenses). All recordings were performed in the home cage. The Microdrive was lowered by 50 μm and then left to stabilise for 3 h and then recordings visualised to determine the presence of single-unit neuronal spiking activity. If single-unit neuronal spiking activity was present, a 3-hour baseline sleep–wake period was recorded (sampling rate 20 kHz). Opto-tagging experiments were performed using blue light (473 nm) stimulation at 10 Hz (10 ms duration) for 1 s, immediately after baseline recording. Stimulation pulses were delivered at least 10 s after the onset of each vigilance states (wake, NREM and REM) as determined by online assessment of the EEG/EMG by an experienced experimenter. Stimulation sessions lasted a maximum of 1 h per day. To determine positive opto-tagging, neuronal spiking activity was first detected and sorted using custom written scripts and activity aligned to laser pulses as recorded by TTL pulses. Neurons that fired within 10 ms of laser pulses with fidelity of >80% were considered to be opto-tagged. The firing characteristics of these opto-tagged neurons across sleep–wake states were then determined from the preceding baseline recording.

**Polysomnographic recordings and state transitions.** All sleep recordings with optogenetic stimulation took place between 12:00 and 15:00 (light onset at 8:00). EOG, EEG and EMG signals derived from the surgically implanted electrodes were amplified (A-M system) and digitised at 512 Hz and digitally filtered and spectrally analysed by fast Fourier transformation using custom-made software (View Point Inc.). Polysomnographic recordings were scored using sleep analysis custom-made software (View Point Inc.). All scoring was performed manually based on the visual signature of the EEG and EMG waveforms, as well as the power spectra of 5-s epochs[45]. We defined wakefulness as desynchronized low-amplitude EEG and high tonic EMG activity with phasic bursts. We defined non-REM sleep as synchronised, high-amplitude, low-frequency (0.5–4 Hz) EEG and highly reduced EMG activity compared with wakefulness with no phasic bursts. We defined REM sleep as having a pronounced theta rhythm (6–9 Hz) and a flat EMG. State transitions were identified when EEG/EMG criterion changes were predominant for >50% of the epoch duration (i.e., 2 s). Polysomnographic scorings were tested by two independent scorers and were found to lie within a 95% confidence interval[45]. The EOG differential signal deflections were used to quantify EMs. We used Spike 2 software to detect EOG signals in an unsupervised manner. Before peak detection, signals were low filter at 1 Hz and downsampled to 125 Hz. Smoothing of the signals and a threshold to detect the peak of the eye deflection were applied. The threshold of the deflections was determined based on the mean amplitude of the total of recording time and was adapted per animal. Low-amplitude deflections were not considered for EM quantification. To determine the response times of EM following optical activation of neurons in the NP$^{Calb}$ or its terminals in the 3N, we determined the times of peak EOG deflection (>200 μV) from the onset of the blue light (473 nm) activation to the first next EM. All detected EMs per animal were aligned to the onset of first light pulse per frequency of stimulation during sleep and wakefulness. Overlay of the average traces per animal of all animals are shown in Figs 3 and 4 and Supplementary Fig. 5. Data collected from all animals were then fitted with a Gaussian curve according to the formula:

$$y = \text{bin.count} \times \exp\left(\frac{(x - \text{mean/s.d.})^2}{2}\right)$$

Median curve values were determined from the apex (maximum $y$ value). All plotting and analysis was performed in Graphpad Prism v.6.0.

**Light stimulation procedures**. In vivo optogenetic activation of the NP$^{Calb}$ was conducted in triplicates. Optical stimulation was delivered using a master-9 pulse generator to deliver 3 mV blue light (473 nm) using 5 ms square pulses at 1 Hz, 10 Hz for 10 s, or 1 s continuous light-on every minute for 1 hour[45]. Note that animals with aberrant circadian distribution of sleep–wake cycle (<10%) were discarded from this study and that these genetic and viral manipulations did not disrupt spontaneous sleep–wake architecture compared to naive and wild-type animals. Animals with the optic fibre above the 3N terminals were activated in a state specific manner, where 1 s continuous light was delivered at the onset of the REM sleep or wake episodes. For the probability, latency and Gaussian distributions analysis, animals with less than two REM sleep episodes in the hour of optogenetic stimulation or with poor EOG signals were not included in the analysis.

The animals for the DTX experiment were injected (AAV9-pCAG-FloxedDTRreverse-WPRE) in the NP$^{Calb}$ as described above. Control animals were wild-type C57/Bl6 mice, non-injected but chronically implanted with EOG/EEG and EMG electrodes. After 2 weeks of implantation animals were habituated to the housing cages and recording cables. After 1 week of habituation, we injected 0.1 ml of saline solution i.p. for 3 consecutive days. On the fourth day we injected DTX (50 μg/kg of body weight) at 11:00 am. Quantification of sleep and EMs were performed on the subsequent hour after injection at the same circadian time. At day 4 post injection, animals were sacrificed for histological verification of the effect of the DTX on DTR-transduced NP$^{Calb}$ neurons.

Inhibition experiments with the silencing construct AAV-EF1a-DIO-eArch3.0-EYFP, were done state specifically where the light was delivered at the onset and throughout the wake or the REM sleep episodes and turned off at the time of transition to a different state. Off-line quantification of EMs was calculated by taking the total number of detected EMs during the light-on periods.

**Statistical analysis**. All behavioural experiments were conducted in at least two cohorts of animals. Statistical significance of comparisons was determined by paired $t$ test (for the experiment consisting of stimulating NP$^{Calb}$ terminals in the 3N), and two-way ANOVA (for NP$^{Calb}$ neuron activation, silencing and DTR-DTX experiment; α adjusted for multiple comparisons), followed by Sidak multiple comparison post hoc test; $P$ values 0.12 ns, 0.33(*), 0.002(**), < 0.001(***) were considered for significance using 0.001 confidence interval. Exact $P$, $t$ and $df$ values are reported in the figure legends.

**Image analysis**. The specimens were evaluated either in a Leica epifluorescence microscope, a Nikon Eclipse fluorescence microscope, a Leica TCS SP5 confocal laser microscope or a Hamamatsu Nanozoomer scanner. Post-processing of the images and contrast adjustments therein were performed using the Adobe Photoshop and Nanozoomer slide-processing softwares.

**Reporting summary**. Further information on research design is available in the Nature Research Reporting Summary linked to this article.

## Data availability

All data generated or analysed during this study are included in this published article (and its supplementary information files).

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

## Acknowledgements

We thank Professor Claudio Bassetti (Bern), for his continuous interest in this project and for promoting and supporting the collaboration between our two research groups. The technical, scientific, and administrative assistance of Simone Eichenberger, Laurence Clément, Christiane Marti, Rachel Ververidis, Ludiwine Aeby, Séverin Rossi, Samara Naim, Michelle Kueffer, Dr. Rosario Arévalo, and Dr. Andrina Zbinden are gratefully acknowledged. Dr A. Babalian performed most of the stereological injections. Dr. S. Arber, (FMI, Basel) kindly supplied several viral constructs, e.g., the DTR. Drs. Andreas Lüthi and Chung Xu (FMI, Basel) furnished us with information and reagents to conduct the experiments with the Rabies virus. This project was funded by the Canton of Fribourg (Switzerland), the Swiss National Foundation (31003A-144036 and 320030-179565 to M.R.C. and 31003-156156 to A.A.), the University of Bern (IRC to C.G.H and A.A.), the Inselspital (to A.A.) and a Consolidator Grant from the European Research Council (to A.A.).

## Author contributions

Conceptualisation: F.G, C.G.H, M.R.C; investigation: F.G, C.G.H, A.B., T.G., D.R.W, M.R.C; resources: A.A, M.R.C; writing-original draft: F.G, C.G.H, A.A., M.R.C; writing-review and editing: F.G, C.G.H, A.A, M.R.C; visualisation: F.G, C.G.H; supervision: F.G, M.R.C; veterinary authorisation: M.R.C, A.A.; funding acquisition: M.R.C, A.A.

## Competing interests

The authors declare no competing interests.
