## [Peer Review File · Nature Communications]

Reviewers' comments:

Reviewer #1 (Remarks to the Author):

REM sleep is a unique phase of sleep in mammals characterized by rapid eye movements (EM). The paper proposes that a population of calbindin-D28K expressing neurons in the nucleus papilio projecting to eye-muscle nuclei is active during REM sleep and plays a pivotal role in REM sleep EM. Although it has some appeal to understand the neural mechanism underlying EM during the eponymous sleep state, I am struggling to understand the significance of the study for the sleep community and the wider field. The authors used sophisticated genetically engineered systems in combination with in-vivo electrophysiology to elucidate EM during REM sleep. This is excellent. However, these techniques are quite standard nowadays and used by many labs around the world. Thus using such techniques does not justify per se publication in Nat Commun. The authors state in the introduction that "we are still puzzled to know whether they (EM) are or are not associated with visual-like activity". Unfortunately, the paper does not make an attempt to address this question or identify other physiological effects of REM sleep EM. Lesioning the NP/CA1b neurons strongly reduces EM during REM sleep, but does not affect sleep/wake behavior. What does it affect? I think this aspect would make the paper significant.

The following comments may help to improve the manuscript:

- a) I do not get the meaning of the statement "P-values <0.01 were considered significant or otherwise stated" on page 21 of the methods section. Apparently, $p < 0.05$ was also considered significant in some figures. Moreover, the authors should show the F and p values for all post hoc tests.
- b) In figure 2e, the groups appear to be significantly different for wake and REM. Please indicate significance. I do not get the point of the statement "We found that the firing activity of opto-tagged NPCalb neurons is modulated across sleep-wake states". The firing activity of many, if not all, neurons in the brain is somehow modulated across sleep/wake states.
- c) The power density of wake and NREM sleep in supplementary figure 6f look similar. Could this indicate a scoring problem?
- d) One "that coordinate" in the second paragraph on page 5 can be deleted.
- e) The manuscript would benefit from professional English editing. For example, the following sentence on page 7 has many weaknesses: "Optical stimulations in freely-moving mice were found to reliably induced action potentials in Chr2-expressing NPCalb neurons firing during REM sleep..." (hyphen between freely and moving, split infinitive, infinitive in past tense, etc.)
- f) The description of AAVs used in the experiments is confusing. For example, the authors use "AAV2/1-EF1alpha-DIO-ChR2-YFP", "ChR2-eYFP" and "AAV2-Ef1a-DIO-eYFP" to describe the same virus. Please unify.
- g) It should be 'rats' in the legend of figure 1b.

Reviewer #2 (Remarks to the Author):

The demonstration provided by the authors that calbindin neurons in the DPGi are specifically responsible for the induction of REM specifically during paradoxical sleep is stunning. It is a key contribution to the identification of the neuronal network responsible for controlling REM sleep. I have only specific comments and requests to improve the manuscript:

In the Introduction, PGO and suspension of homeostasis should be removed since PGO is not present in mice and homeostasis is too vague.

Page 3, the authors state that PGO waves are called P-waves in rats. However, it has not been clearly shown that they correspond to the same waves. Indeed cats PGO are generated by cholinergic neurons of PPTg and Ldtg that project to the lateral geniculate. P waves have been only

recorded in the rat SLD region. The neurons responsible for their genesis have not been identified. More work is needed to confirm that they correspond to the same waves.

In the result section what means $c-fos > 0$? In $c-fos > 0$ $Calb > 0$?

What is the percentage of $c-fos$ labeled neurons not expressing calbindin? This information is missing and is important to give an idea of the contribution of the $Calb$ neurons to the $c-fos$ labeled neurons.

Further, the authors do not report whether the calbindin neurons are localized in the rostral, intermediate or caudal part of the DPGi. They should look to their sections stained with $cFos$ and determine whether the $Calb$ neurons are in their large majority located rostrally.

Indeed Sapin et al. (2009) showed that only the most rostral DPGi neurons labeled with $cFos$ after paradoxical sleep rebound were not GABAergic:

<https://journals.plos.org/plosone/article/file?type=supplementary&id=info:doi/10.1371/journal.pone.0004272.s004>

It seems therefore that the authors are studying the rostral neurons which are indeed glutamatergic. It means that you have two radically different types of neurons in the DPGi, the glutamate involved in REM genesis and the GABA in the inhibition of waking neurons particularly from the LC. The authors should refer to and discuss this point.

The authors show that the $Calb$ neurons recorded are more active during W and REM. They however do not report whether their activity during waking is linked with EM and should. Indeed, this is important to understand their function. In Fig. 3C, they also show that the neurons are very active during Waking compared to REM. It however seems rather tonic. Maybe the discharge mode is key to understand the function.

Page 5 remove repetition in the sentence:

Anterograde mapping (Table 1) revealed a large number of axonal endings in the contralateral nuclei that coordinate that co-ordinate the EMs

MdV is ranked in Table 1 in the structures implicated in REM sleep control. I'm not aware that this is the case.

Supplementary Fig. 2, the red anterograde labeling should be enhanced a bit to be more clearly seen.

Reviewer #3 (Remarks to the Author):

This is an extremely impressive technical achievement that addresses the origin of eye movements during REM sleep. This issue has been examined before, with a particularly relevant investigation being [1]. This paper showed a similar loss of REM eye movements (and PGO spikes), with preservation of REM sleep of normal duration from a very different lateral pontine lesion. The authors should reference and discuss the possible relationship of these two critical areas.

An additional consideration that should be discussed is the transection work that most convincingly localizes the REM sleep (and rapid eye movement) generation region. Depending on the precise transection level, REM sleep with PGO waves, rapid eye movements and atonia can be seen after removing forebrain regions and disconnecting medullary regions (with the exception of atonia). The dorsal paragigantocellularis nucleus that is the focus of the present work appears to be within this critical regions (in the cat) – but just barely, at the most caudal limit. Barely may be enough and this should be mentioned with reference to the critical publications[2-5]. Of course, because of species differences in brainstem anatomy as well as the angle of transection this is not crystal clear in these mouse studies.

Reference List

1. Shouse M.N. and Siegel J.M. (1992). Pontine regulation of REM sleep components in cats:

integrity of the pedunculo-pontine tegmentum (PPT) is important for phasic events but unnecessary for atonia during REM sleep. *Brain Res* 571: 50-63.

2. Jouvet M. (1962). Recherches sur les structures nerveuses et les mecanismes responsables des differentes phases du sommeil physiologique. *Arch. ital. Biol.* 100: 125-206.

3. Siegel J.M., Nienhuis R., and Tomaszewski K.S. (1984). REM sleep signs rostral to chronic transections at the pontomedullary junction. *Neurosci. Lett.* 45: 241-246.

4. Siegel J.M., Tomaszewski K.S., and Nienhuis R. (1986). Behavioral states in the chronic medullary and mid-pontine cat. *Electroenceph. Clin. Neurophysiol.* 63: 274-288.

5. Webster H.H. and Jones B.E. (1988). Neurotoxic lesions of the dorsolateral pontomesencephalic tegmentum-cholinergic cell area in the cat II. Effects upon sleep-waking states. *Brain Res.* 458: 285-302.

Reviewers' comments

Our answers to reviewer's comments

Reviewer #1 (Remarks to the Author):

REM sleep is a unique phase of sleep in mammals characterized by rapid eye movements (EM). The paper proposes that a population of calbindin-D28K expressing neurons in the nucleus papilio projecting to eye-muscle nuclei is active during REM sleep and plays a pivotal role in REM sleep EM. Although it has some appeal to understand the neural mechanism underlying EM during the eponymous sleep state, I am struggling to understand the significance of the study for the sleep community and the wider field. The authors used sophisticated genetically engineered systems in combination with in-vivo electrophysiology to elucidate EM during REM sleep. This is excellent. However, these techniques are quite standard nowadays and used by many labs around the world. Thus using such techniques does not justify per se publication in Nat Commun. The authors state in the introduction that “we are still puzzled to know whether they (EM) are or are not associated with visual-like activity”. Unfortunately, the paper does not make an attempt to address this question or identify other physiological effects of REM sleep EM. Lesioning the NP/CALb neurons strongly reduces EM during REM sleep, but does not affect sleep/wake behavior. What does it affect? I think this aspect would make the paper significant.

We agree that an insight into the function(s) of EMs during REM sleep would be of great scientific interest. However, a tackling of this question transcends the scope of the present study. Using state-of-the-art techniques in the field of neuroscience, we report on the neural mechanisms that govern the generation of EMs during REM sleep. The findings alone merit publication. “*The capability to induce EMs during REM sleep on command affords a powerful tool for the investigation of their functions*”. This sentence now brings the text to a close in the manuscript (page 11). The Swiss National Foundation has granted us a financial support for 3 years to expand the studies that are reported in the submitted manuscript.

The following comments may help to improve the manuscript:

a) I do not get the meaning of the statement “P-values <0.01 were considered significant or otherwise stated” on page 21 of the methods section. Apparently, $p < 0.05$ was also considered significant in some figures. Moreover, the authors should show the F and p values for all post hoc tests.

We have now corrected the statement on page 21 to “*P-values 0.12ns, 0.33(*), 0.002(**), <0.001(***) were considered for significance using 0.001 confidence interval. Exact P, t and df values are reported in the figure legends*”; and we included all the F and P values throughout the text.

b) In figure 2e, the groups appear to be significantly different for wake and REM. Please indicate significance. I do not get the point of the statement “We found that the firing activity of opto-tagged NPCalb neurons is modulated across sleep-wake states”. The firing activity of many, if not all, neurons in the brain is somehow modulated across sleep/wake states.

Statistical values and significance have now been corrected in Fig 2e and the figure legend. We thank the reviewer for rendering us attentive to the lack of clarity in the statement quoted above. We have now corrected it to “*During REM sleep, the firing activity of the*

opto-tagged NP^{Calb} neurons was augmented relative to that in other cells of the NP area (viz., non-responders) (REM: 6.9 ± 0.5 Hz in responder cells versus 0.025 ± 0.0572 in non-responders; Fig. 2e)." on pages 4-5.

c) The power density of wake and NREM sleep in supplementary figure 6f look similar. Could this indicate a scoring problem?

We believe that this is due to the time window used. To clarify our point, we have now included in this figure representative power densities, EEG and EM of sleep/wake episodes using longer time window before and after the stimulation paradigm.

d) One "that coordinate" in the second paragraph on page 5 can be deleted.

This has now been corrected.

e) The manuscript would benefit from professional English editing. For example, the following sentence on page 7 has many weaknesses:

"Optical stimulations in freely-moving mice were found to reliably induced action potentials in ChR2-expressing NPCalb neurons firing during REM sleep..." (hyphen between freely and moving, split infinitive, infinitive in past tense, etc.)

The manuscript has been reviewed and corrected by a professional English editor. Nevertheless, few sentences were added to the manuscript after this correction, amongst which was the one described by reviewer 1. We have now corrected these sentences.

f) The description of AAVs used in the experiments is confusing. For example, the authors use "AAV2/1-EF1alpha-DIO-ChR2-YFP", "ChR2-eYFP" and "AAV2-Ef1a-DIO-eYFP" to describe the same virus. Please unify.

We thank the reviewer for rendering us attentive to this lack of consistency. AAV2/1-EF1alpha-DIO-hChR2(H134R)-YFP (ChR2) and AAV2/1-EF1alpha-DIO-EYFP (YFP; control) are the complete names of the virus described in the Methods section. Animals injected with these virus are now consistently referred as to "ChR2 or control" throughout the manuscript.

g) It should be 'rats' in the legend of figure 1b.

No, it is correct: Fig1b is mouse, Fig1c is rat, as stated in the legend to Fig1.

Reviewer #2 (Remarks to the Author):

The demonstration provided by the authors that calbindin neurons in the DPGi are specifically responsible for the induction of REM specifically during paradoxical sleep is stunning. It is a key contribution to the identification of the neuronal network responsible for controlling REM sleep.

We thank the Reviewer for the supportive comments.

I have only specific comments and requests to improve the manuscript:

- In the Introduction, PGO and suspension of homeostasis should be removed since PGO is not present in mice and homeostasis is too vague.

We corrected the manuscript according to the Reviewer's concerns.

Page 3, the authors state that PGO waves are called P-waves in rats. However, it has not been clearly shown that they correspond to the same waves. Indeed cats PGO are generated by cholinergic neurons of PPTg and Ldtg that project to the lateral geniculate. P waves have been only recorded in the rat SLD region. The neurons responsible for their genesis have not been identified. More work is needed to confirm that they correspond to the same waves.

We thank the Reviewer for this clarification. We previously assumed that both PGO and P waves were identical since they were recorded from adjacent brain structures, though in different species. Thus, the original sentence "endogenous signals that in rats are called P-waves" is now replaced in page 3 by "*which might correspond to the P-waves in rats*".

- In the result section what means >0? In c-fos>0 Calb>0 ?

On page 4, we changed the original sentence "(46.5 ± 9.3 % and 42.2 ± 5 % of double c-fos>0 Calb>0, respectively; Fig. 2a-c)" to "*(the proportion of Calb-immunoreactive neurons expressing c-fos being 46.5 ± 9.3 % in rats and 42.2 ± 5 % in mice; Fig. 2a-c)*".

What is the percentage of c-fos labeled neurons not expressing calbindin? This information is missing and is important to give an idea of the contribution of the Calb neurons to the cfos labeled neurons.

In horizontal sections, the NP^{Calb} area can be precisely delimited by Calb immunoreactivity, from the rostral to the caudal part of the DPGi. The percentage of c-fos labelled neurons that do not express Calb is 53.35 ± 11.89 % (n=4 animals) in the REM rebound group. This information is now included in the text on page 4: "*Furthermore, 53.35 ± 11.89 % (n=4 rats) of the c-fos-labelled neurons in the NP^{Calb} failed to express Calb.*"

- Further, the authors do not report whether the calbindin neurons are localized in the rostral, intermediate or caudal part of the DPGi. They should look to their sections stained with cFos and determine whether the Calb neurons are in their large majority located rostrally. Indeed Sapin et al. (2009) showed that only the most rostral DPGi neurons labeled with cFos after paradoxical sleep rebound were not GABAergic:

<https://journals.plos.org/plosone/article/file?type=supplementary&id=info:doi/10.1371/journal.pone.0004272.s004>

It seems therefore that the authors are studying the rostral neurons which are indeed glutamatergic. It means that you have two radically different types of neurons in the DPGi, the glutamate involved in REM genesis and the GABA in the inhibition of waking neurons particularly from the LC. The authors should refer to

and discuss this point.

We agree with Reviewer 2 that the DPGi contains “two radically different types of neurons” (this phrase has been introduced in the discussion, page 9). However, the glutamatergic NP^{Calb} are distributed from the rostral to the most caudal part of the DPGi.

Sapin and collaborators (Sapin et al. 2019) found that within the rat DPGi, the percentage of GAD/c-fos positive neurons was higher in the caudal part of the DPGi (86%) than in the rostral part (39%). As presented in the horizontal rat brain section of figure 2b, no evidence for such a rostro-caudal difference is discernible for Calb/c-fos immunoreactivity (see the enclosed image). After re-examination of the mouse brain sections, we observe even in this species an evenly distributed Calb/c-fos neurons in the whole DPGi. This difference between the GABAergic and glutamatergic DPGi neurons was already mentioned in our original manuscript (page 4, Results Section). This is now further discussed in the Discussion section (Page 9): “*The DGPi harbours two radically different types of neuron: the glutamatergic NP^{Calb} neurons described here, and the GABAergic ones¹⁶⁻²⁰. While the latter are involved in the inhibition of waking neurons, particularly those from the LC, the NP^{Calb} neurons project excitatory connections to the nuclei of the external eye muscles*”.

And on page 11: “*Although no significant effects on the onset and the maintenance of REM-sleep parameters were observed either upon the activation, the inhibition or the deletion of NP^{Calb} neurons, further studies are required to ascertain whether the competence of these neurons - or of others intermingling Calb-negative ones in this region - transcend in scope its premotoric role in the generation of EMs during REM sleep*”.

-The authors show that the Calb neurons recorded are more active during W and REM. They however do not report whether their activity during waking is linked with EM and should. Indeed, this is important to understand their function. In Fig. 3C, they also show that the neurons are very active during Waking compared to

REM. It however seems rather tonic. May be the discharge mode is key to understand the function.

Indeed, the discharge rate of NP^{Calb} neurons is higher during wakefulness; however, we did not find any change in their mode of discharge across states. During wake, their high activity, and constant EMs (probably due to visual experience), make it difficult to time lock those events, as we could do for REM sleep where EMs represent single muscular events (and not a continuous fluctuating signals).

- Page 5 remove repetition in the sentence:

Anterograde mapping (Table 1) revealed a large number of axonal endings in the contralateral nuclei that coordinate that co-ordinate the EMs

This has now been corrected.

- MdV is ranked in Table 1 in the structures implicated in REM sleep control. I'm not aware that this is the case.

We fully agree with this remark, and we have to admit that we confused MdV and the ventral medulla neurons described by Weber and Dan 2015 Nature 526:435-438. This is now corrected in the new version: MdV is not anymore presented (both in the Text and in Table 1) as a structure implicated in REM sleep control.

- Supplementary Fig. 2, the red anterograde labeling should be enhanced a bit to be more clearly seen.

This has now been done.

Reviewer #3 (Remarks to the Author):

This is an extremely impressive technical achievement that addresses the origin of eye movements during REM sleep. This issue has been examined before, with a particularly relevant investigation being [1]. This paper showed a similar loss of REM eye movements (and PGO spikes), with preservation of REM sleep of normal duration from a very different lateral pontine lesion. The authors should reference and discuss the possible relationship of these two critical areas.

An additional consideration that should be discussed is the transection work that most convincingly localizes the REM sleep (and rapid eye movement) generation region. Depending on the precise transection level, REM sleep with PGO waves, rapid eye movements and atonia can be seen after removing forebrain regions and disconnecting medullary regions (with the exception of atonia). The dorsal paragigantocellularis nucleus that is the focus of the present work appears to be within this critical regions (in the cat) – but just barely, at the most caudal limit. Barely may be enough and this should be mentioned with reference to the critical publications[2-5]. Of course, because of species differences in brainstem anatomy as well as the angle of transection this is not crystal clear in these mouse studies.

We thank Reviewer #3 for his/her supportive comments.

We have carefully read the four papers on transectioning and lesioning experiments that are alluded to by the Reviewer. In the publication by Shouse et al. (1992) and Webster et al. (1988) electrolytic and radio-frequency, respectively kainic acid induced lesioning of the PPT led to a reduction of EMs and PGO-waves during REM sleep. These interventions eliminated both the neurons as well as axons. In Fig 3 (A1, APO and P1) in the publication by Shouse et al (1992), lesioning in one of the brain (striped drawing) affected also the IIIrd and eventually also the IVth eye-muscle nuclei in the midbrain, namely the targets of the projection from the *NPC^{Calb}* neurons in the medulla oblongata. In Fig. 1 of the publication by Webster et al (1988), the lesion in the distal pons occupies a region in which the axons of the *NPC^{Calb}* projection run. Since the specificity of kainic acid in killing neuronal cell bodies has been questioned several times in the past (e.g., [science.sciencemag.org > content > 1417.1.full.pdf](http://science.sciencemag.org/content/1417.1.full.pdf)), also the axons could have been damaged. In other words, the lesions in these two former publications may have eliminated the target of our projection, or interrupted its axonal path. Therefore, in the revised manuscript the following paragraph has been inserted (page 9):

*“The data gleaned from investigations using different approaches, namely transectioning, lesioning or chemical inhibition, have established the pons to be the brain region of the brain responsible for the evocation of REM sleep, EMs, PGO-spikes and atonia (1,8,26-31). For example, lesioning of the pedunculo-pontine tegmentum lead to a dramatic reduction in the number of EMs and of PGO-spikes (30). However, some of these pontine lesions may have intersected the pons-traversing pathway from the *NPC^{Calb}* in the upper medulla to the nuclei of the external eye muscles in the midbrain”.*

Concerning the transection work, we write:

“The dorsal paragigantocellular nucleus (DPGi), which was the focus of the present investigation, appears to be situated at the most caudal limit of this critical pontine region”.

Reference List

1. Shouse M.N. and Siegel J.M. (1992). Pontine regulation of REM sleep components in cats: integrity of the pedunculo-pontine tegmentum (PPT) is important for phasic events but unnecessary for atonia during REM sleep. *Brain Res* 571: 50-63.
2. Jouvet M. (1962). Recherches sur les structures nerveuses et les mecanismes responsables des differentes phases du sommeil physiologique. *Arch. ital. Biol.* 100: 125-206.
3. Siegel J.M., Nienhuis R., and Tomaszewski K.S. (1984). REM sleep signs rostral to chronic transections at the pontomedullary junction. *Neurosci. Lett.* 45: 241-246.
4. Siegel J.M., Tomaszewski K.S., and Nienhuis R. (1986). Behavioral states in the chronic medullary and mid-pontine cat. *Electroenceph. Clin. Neurophysiol.* 63: 274-288.
5. Webster H.H. and Jones B.E. (1988). Neurotoxic lesions of the dorsolateral pontomesencephalic tegmentum-cholinergic cell area in the cat II. Effects upon sleep-waking states. *Brain Res.* 458: 285-302.

REVIEWERS' COMMENTS:

Reviewer #1 (Remarks to the Author):

The authors have sufficiently addressed/corrected my minor comments.

My major criticism regarding the physiological function of EMs remains unsatisfied. In light of the comments of the other reviewers and challenging task to address this question, I feel that the novel findings and the paper's technical level may justify publication.

Reviewer #2 (Remarks to the Author):

Thank you for perfectly answering to my questions and requests. I have no more comments on your manuscript.